# A Rapid Alkalinization Factor-like Peptide EaF82 Impairs Tapetum Degeneration during Pollen Development through Induced ATP Deficiency

**DOI:** 10.3390/cells12111542

**Published:** 2023-06-04

**Authors:** Chiu-Yueh Hung, Farooqahmed S. Kittur, Keely N. Wharton, Makendra L. Umstead, D’Shawna B. Burwell, Martinique Thomas, Qi Qi, Jianhui Zhang, Carla E. Oldham, Kent O. Burkey, Jianjun Chen, Jiahua Xie

**Affiliations:** 1Department of Pharmaceutical Sciences, Biomanufacturing Research Institute & Technology Enterprise, North Carolina Central University, Durham, NC 27707, USA; chiuyueh@gmail.com (C.-Y.H.); fkittur@nccu.edu (F.S.K.); jianhui.zhang@montana.edu (J.Z.); coldham@nccu.edu (C.E.O.); 2USDA-ARS Plant Science Research Unit and Department of Crop and Soil Sciences, North Carolina State University, Raleigh, NC 27695, USA; kent.burkey@ars.usda.gov; 3Mid-Florida Research and Education Center, Environmental Horticulture Department, Institute of Food and Agricultural Sciences, University of Florida, Apopka, FL 32703, USA

**Keywords:** ATP deficiency, cysteine-rich peptide, *Epipremnum aureum*, kinase AKIN10, pollen production, rapid alkalinization factor, tapetum degeneration

## Abstract

In plants, the timely degeneration of tapetal cells is essential for providing nutrients and other substances to support pollen development. Rapid alkalinization factors (RALFs) are small, cysteine-rich peptides known to be involved in various aspects of plant development and growth, as well as defense against biotic and abiotic stresses. However, the functions of most of them remain unknown, while no RALF has been reported to involve tapetum degeneration. In this study, we demonstrated that a novel cysteine-rich peptide, EaF82, isolated from shy-flowering ‘Golden Pothos’ (*Epipremnum aureum*) plants, is a RALF-like peptide and displays alkalinizing activity. Its heterologous expression in Arabidopsis delayed tapetum degeneration and reduced pollen production and seed yields. RNAseq, RT-qPCR, and biochemical analyses showed that overexpression of *EaF82* downregulated a group of genes involved in pH changes, cell wall modifications, tapetum degeneration, and pollen maturation, as well as seven endogenous Arabidopsis *RALF* genes, and decreased proteasome activity and ATP levels. Yeast two-hybrid screening identified AKIN10, a subunit of energy-sensing SnRK1 kinase, as its interacting partner. Our study reveals a possible regulatory role for RALF peptide in tapetum degeneration and suggests that EaF82 action may be mediated through AKIN10 leading to the alteration of transcriptome and energy metabolism, thereby causing ATP deficiency and impairing pollen development.

## 1. Introduction

Since the discovery of systemin in 1991 [1], numerous plant peptide hormone-like sequences have been identified [2], and the quest for understanding their functions and action modes is growing. One such peptide family is rapid alkalinization factors (RALFs), which are small cysteine-rich peptides (CRPs) widely present in the plant kingdom [3]. Many RALF-like sequences have been identified from genomic sequences [4,5] and a phylogenetic analysis of 795 RALFs from 51 plant species showed that they fall into four distinct clades [5]. However, only a handful of RALFs belonging to clades I, II, and III have been functionally characterized and shown to be involved in cell expansion, root and root hair development, pollen tube elongation/rupture, and immunity, while the majority of them have not been studied functionally [6]. The clade IV is the least investigated and distinct from other clades with greater evolutionary divergence [5,6].

Studies related to the function and mechanism of RALFs are attracting greater attention due to their importance in plant growth and development, and defense against biotic and abiotic stresses [7,8]. Recent studies on Arabidopsis AtRALF1/23/4/19 demonstrated that their actions are perceived through interactions with *Catharanthus roseus* Receptor Like Kinase 1 Like (CrRLK1L) receptors [6]. Binding of AtRALFs to CrRLK1L receptors triggers downstream signal transduction, including downregulation of H^+^-ATPase resulting in extracellular pH increase. The increased extracellular pH was thought to strengthen the cell wall to prevent pathogen invasion and make alkaline apoplasts an unfavorable environment for pathogens [9]. The pH increase also counteracts the acidification-induced loosening of the cell wall, leading to inhibition of root growth and pollen tube elongation [10]. As for clade IV RALFs, many members are thought to be involved in pollen development based on their transcriptional patterns. Studies on male sterile Chinese cabbage lines found that three out of fourteen identified differentially expressed *RALFs* belong to clade IV [11]. Studies on four pollenless Arabidopsis tapetum mutants also revealed that five out of seven differentially expressed *RALFs* are members of clade IV [12]. In Arabidopsis, six out of eight pollen-abundant *AtRALFs* (4/8/9/15/19/25/26/30) were found to belong to clade IV [13]. However, to date, their precise roles and underlying mechanism of action in pollen development are still unknown. 

We previously isolated a novel gene, *EaF82*, from shy-flowering ‘Golden Pothos’ plants (*Epipremnum aureum*) [14,15]. The gene encodes a 120 amino acid CRP whose highest match peptide at that time was MiAMP1, an antimicrobial peptide from *Macadamia integrifolia* [16]. ‘Golden Pothos’ is one of the most popularly grown houseplants worldwide due to its variegated leaves and tolerance to low light. It can be used for studying chloroplast biogenesis by comparing genes in the cells between color defective and normal green sectors [14,16]. *EaF82* was found to be expressed in all types of vegetative tissues, such as leaves, petioles, nodes, and roots at different levels [16]. Within a single leaf, it was differentially expressed between green and yellow/white sectors of variegated ‘Golden Pothos’ leaves [14,16], and the expression was positively regulated by auxin [16]. However, the function of EaF82 remained largely obscure. 

In the present study, we aimed to further elucidate its function. We utilized its peptide sequence similarity analysis and found that EaF82 is closely related to clade IV RALFs, and that the synthetic EaF82 peptide could induce alkalinization of tobacco suspension cell culture medium as typical RALFs do. Since the clade IV RALFs might be involved in the pollen development [11,12,13], it is essential to investigate whether EaF82 plays any role in floral organs/tissues. Due to the challenge in elucidating its potential function in the reproductive organs/tissues in the shy-flowering ‘Golden Pothos’ plants, we adopted a gain-of-function approach commonly used for studying genes in nonmodel, long life cycle species (e.g., poplar) [17] and ectopically expressed *EaF82* in the model species Arabidopsis (*Arabidopsis thaliana*) and tobacco (*Nicotiana benthamiana*). Overexpressing *EaF82* in Arabidopsis plants was found to affect pollen production and seed setting associated with induced transcriptomic changes and decreased ATP levels in flowers. Furthermore, we identified AKIN10, a catalytic α-subunit of energy sensor—sucrose nonfermenting related kinase 1 (SnRK1)—as an interacting partner of EaF82. We discuss its potential action mechanism through binding to AKIN10 to alter transcriptome and energy metabolism, leading to impaired tapetum degeneration and pollen development.

## 2. Materials and Methods 

### 2.1. Construction of Genetic Cassettes

Three new genetic cassettes were created in this study based on the pBI121 vector: *EaF82p::EaF82-sGFP*, *35Sp::EaF82-sGFP*, and *35Sp::EaF82* (designated as TA, TB and TC) (Appendix A). The genetic cassette *EaF82p::GUS* was created in a previous study [16]. The vector pBI121 carrying the *CaMV35S* promoter (*35Sp*) driving bacterial *uidA* (*GUS*) (*35Sp::GUS*) was used as a control (designated as C). To create these genetic cassettes, the intronless gene *EaF82*, previously isolated from *E. aureum* containing a 1.2 kb *EaF82* promoter and a 0.8 kb full length *EaF82* including *EaF82* terminator [16], was used. The two previously constructed plasmid DNAs, CEJ826 and CEJ937 (CEJ # is a nomenclature for plasmid DNA in Xie’s laboratory), and the pBI121 vector were used to construct these genetic cassettes. The plasmid DNA CEJ826 contains the *EaF82* promoter (*EaF82p*) and full length *EaF82* including *EaF82* terminator in pCR^®^-blunt 3.5 kb vector (Invitrogen, Waltham, MA, USA). The plasmid DNA CEJ937 contains *35Sp* and 0.357 kb *EaF82* coding region without TGA stop codon in-frame fused with *sGFP* in pUC19 (Invitrogen). To construct the TC genetic cassette, the *EaF82* fragment isolated from CEJ826 by EcoRI-blunted and EcoRV digestion was ligated to vector pBI121 predigested by SacI-blunted and SmaI. The resultant contained *35Sp* driving full length *EaF82* including *EaF82* terminator. To construct the TB genetic cassette, the *35Sp::EaF82-sGFP-NosT* fragment isolated from CEJ937 by HindIII and EcoRI digestion was ligated to pBI121 predigested by HindIII and EcoRI. The resultant plasmid DNA was named CEJ964. To construct the TA genetic cassette, the *EaF82* promoter fragment isolated from CEJ826 by XbaI and EcoRV digestion was ligated to predigested CEJ964 by HindIII-blunted and XbaI. The resultant construct TA had *EaF82-sGFP* driven by *EaF82p*, whereas TB had the same *EaF82-sGFP* but driven by *35Sp*. The *Agrobacterium* strain LBA4404 was used to harbor these constructs.

### 2.2. Created Transgenic Plants

Transgenic plants were generated with *A. thaliana* Col-0, *N. benthamiana* tobacco using previously described transformation methods [16,18] except the kanamycin concentration for selection in *N. benthamiana* was 300 mg/L. The Arabidopsis *EaF82p::GUS* transgenic plants were created previously [16]. All experiments were performed using homozygous lines. 

### 2.3. Alkalinization Assay

The alkalinization assay was performed with tobacco (*N. tabacum*, W38) suspension-cultured cells (Appendix A) derived from leaf calli, following the method described by Pearce et al. [7]. The EaF82-S peptide (Figure 1a) was synthesized by Lifetein (Somerset, NJ, USA) with a purity of 94.55%. Its molecular weight was 10 kD as confirmed using an Electrospray Ionization (ESI) Mass Spectrometry. Alkylated EaF82-S peptide, prepared as previously reported [7], was used as a negative control. Initially, 70 mL of suspension cells (OD_600_ = 0.2) in liquid medium, pH 5.6, containing MS basal salts and vitamins (Research Product International, Mount Prospect, IL, USA), were grown in a 300 mL flask under dark at 25 °C with shaking at 160 rpm until the cell density reached 5–8 × 10^4^ cluster cells/mL after 3–4 days of subculture. For the assay, each 10 mL cell suspension was aliquoted into a 6-well plate and acclimated with shaking for 2–3 h. The peptide was reconstituted in ddH_2_O to a concentration of 100 µM as a stock solution. Peptide concentrations of 1, 5, 10, 50, and 100 nM were used for the assay. The pH was measured using a Mettler Toledo^TM^ InLab 413 pH Electrodes (Mettler-Toledo).

### 2.4. RT-PCR and RT-qPCR

The procedures for RNA isolation, RT-PCR, and RT-qPCR were the same as described in Hung et al. [14]. Data from three sets of biological samples were averaged. The primers are listed in Appendix A. The QuantumRNA™ Universal 18S Internal Standard (Invitrogen) was used as an internal control.

### 2.5. RNAseq and DEG Analyses

The Illumina sequencing was performed by DNA Link (San Diego, CA, USA). The sequencing reactions were run on the Illumina NextSeq 500 with single-end 76 bp read. The Consensus Assessment of Sequence and Variation (CASAVA) software version 1.8.2 (Illumina) was used to remove adaptor sequences, nucleotide library indexes, and generate fastq files. The RNAseq reads were mapped to the *A. thaliana* reference genome TAIR10 using TopHat [19] to produce aligned reads and FPKM [20]. The gene annotations were based on NCBI database. The DEG analysis was conducted by Cufflinks and Cuffdiff [19]. The PANTHER classification system was used for functional pathway analysis [21].

### 2.6. Protein Isolation, SDS-PAGE and Immunoblotting

For extracting total proteins from leaves and seedlings, the Plant Total Protein Extraction Kit (Sigma-Aldrich, St. Louis, MO, USA) was used. To extract pollen proteins, the method from Chang and Huang [22] was adopted. In brief, hundreds of opened flowers were harvested in an ice-cold tube and about 5× volume of cold extraction buffer (HEPES potassium solution) was added. After brief mixing, the mixture was centrifuged at 350× *g* for 1 min to collect pollen. The purity of collected pollen was examined under a microscope (Appendix A) before disruption using a blue pestle on ice. After a 1 h incubation on ice, extracted pollen proteins in the supernatant were collected following centrifugation at 18,000× *g* for 30 min at 4 °C. To extract proteins from unopened flower clusters, tissues ground in liquid nitrogen were mixed in a 1:2.5 ratio with extraction buffer containing 50 mM HEPES pH 7.8, 2 mM EDTA, 1 mM DTT and 1× Halt^TM^ Protease Inhibitor Cocktail, EDTA-free (Thermo Scientific, Waltham, MA, USA). After centrifugation at 18,000× *g* for 20 min at 4 °C, the protein extracts were collected.

SDS-PAGE, immunoblotting, and the subsequent detection of chemiluminescent signals, along with staining of blots, were performed as previously described [16]. To detect EaF82, a custom-made anti-EaF82 antibody was used [16]. To detect GFP, the blots were probed with 1:200 diluted mouse monoclonal anti-GFP (sc-9996, Santa Cruz Biotechnology, Dallas, TX, USA). After three washes with TBST, the blots were incubated with 1:10,000 diluted HRP-conjugated anti-mouse IgGκ (sc-516102, Santa Cruz Biotechnology). For detecting AKIN10, the blots were probed with 1:500 diluted anti-AKIN10 (AS10919, Agrisera, Vännäs, Sweden) in PBST containing 1% dry milk, followed by the 1:20,000 diluted HRP-conjugated anti-rabbit IgG (H + L) (AS014, ABclonal, Woburn, MA, USA). For quantification of band intensity, the same blot was probed with 1:15,000 diluted plant actin mouse mAb (AC009, ABclonal) as an internal control, followed by the 1:20,000 diluted HRP-conjugated anti-mouse IgG (Jackson ImmunoResearch, West Grove, PA, USA). Three independent experiments were performed and scanned images of band intensities on X-ray films were quantified using Image J (https://imagej.nih.gov/ij/) (accessed on 9 October 2020). To examine the proteasome subunits, the same procedure used for detecting AKIN10 was also used, except for the incubation conditions for each primary antibody. Anti-RPN6 (AS15 2832A, Agrisera) was 1:1000 diluted in TBST containing 3% dry milk and incubated at 23 °C for 1 h. Anti-RPN10 (PHY0102S, PhytoAB, San Jose, CA, USA) was 1:2000 diluted in TBST and incubated at 23 °C for 1 h. Anti-Rpt5a/b (PHY1747A, PhytoAB) was 1:1000 diluted in PBST and incubated at 4 °C for 16 h.

To determine the size of EaF82, the Novex™ tricine gel system (Invitrogen) for revealing low molecular weight proteins was used. In brief, extracted proteins were first mixed with equal volume of tricine SDS sample buffer containing 50 mM DTT. Before loading onto 16% tricine gel, the sample mixtures were denatured at 85 °C for 2 min. The Spectra™ Multicolor Low Range Protein Ladder, a mixture of six proteins ranging from 1.7 to 40 kD, was used as size standards. Proteins were separated in tricine SDS running buffer under constant 50 volts for 4 h and later increasing to 100 volts for 1 h. They were then transferred to 0.2 μm PVDF membrane in Novex Tris-Glycine transfer buffer with 20% methanol under constant 20 volts for 90 min. Immunoblotting was performed as described for detecting EaF82. 

### 2.7. Histological Analysis and Seed Counting

GUS assay was performed as previously described [16]. To detect GFP, tissues were observed directly under a fluorescence stereo microscope Nikon SMZ1000 (Nikon) equipped with an ET-Narrow Band EGFP to minimize autofluorescence ex 480 nm/20 and em 510 nm/20 (49020, Chroma Technology, Bellows Falls, VT, USA). The images were captured using a Nikon Digital Sight DS-Fi1 and analyzed using software NIS-ELEMENTS BR 3.0 (Nikon). To observe pollen, pollen grains were released by gently crushing the anthers on slides. For observing pollen germination, the method described by Krishnakumar and Oppenheimer [23] was used to prepare slides containing germinated pollen grains. The pollen images were observed under a Zeiss LSM 710 microscope with ZEN 2011 Imaging software (Zeiss). For pollen viability assay, the stamens were tapped on a drop of iodine/potassium iodide TS 1 solution (RICCA chemical company) placed on a glass slide. After 5 min in dark, the pollen grains were imaged under a Keyence BZ-X700 microscope (Keyence). To observe pollen development, the whole unopened flower cluster was fixed in FAA solution (4% [*v/v*] formaldehyde, 5% [*v/v*] acetic acid, and 50% [*v/v*] ethanol) with gentle vacuum for 5 min and then kept at 4 °C for 24 h. The dehydration, paraffin embedding, and sectioning steps were the same as described by Hung et al. [15]. After immobilizing on slides, the deparaffinized specimens were stained for 2–3 min with fresh 0.05% (*w/v*) Toluidine blue O solution in 0.1 M citrate phosphate buffer, pH 6.8, and then rinsed in running water. After mounting on Fluoromount (Sigma-Aldrich), the images were captured using a Keyence BZ-X700. For counting seeds, images of seeds were counted using Visionworksls Software 5 (UVP). 

### 2.8. ATP Measurement

The method of ATP measurement from Napolitano and Shain [24] was adopted. In brief, ATP was extracted by mixing 1 mg of tissue powder with 40 µL of 50 mM HEPES buffer, pH 7.4, containing 33 mAU/mL Novagen^®^ proteinase K (Millipore, Burlington, MA, USA). The mixture was incubated at 50 °C for 15 min, then at 80 °C for 5 min. After centrifugation at 20,000× *g* for 10 min at 4 °C, the supernatant was collected and used for ATP assay with an adenosine 5-triphosphate (ATP) bioluminescent assay kit (Sigma-Aldrich). The generated bioluminescence signal was measured using a SpectraMax M5 plate reader (Molecular Devices).

### 2.9. Proteasomal Activity Assay

Proteasomal activity was assayed as described by Vallentine et al. [25], except that 10 mM ATP and 5% (*v/v*) glycerol were included in the extraction buffer but omitted in the reaction buffer. Each assay had equal amounts of extracted proteins (3 µg). The generated signal was measured using a microplate reader PHERAstar (BMG Labtech).

### 2.10. Yeast Two-Hybrid

The Y2H screening and 1-by-1 direct interaction assays were conducted by Hybrigenics Services SAS (Paris, France). The coding sequence for EaF82 (amino acid residues 30-120) was PCR-amplified from previously constructed CEJ982 containing *EaF82* and cloned in frame with the LexA DNA binding domain (DBD) into pB27 vector as a *C*-terminal fusion to LexA (N-LexA-EaF82-C). Hybrigenics’ reference for this construct was hgx4998v1_pB27. The entire insert sequence in the construct was confirmed by sequencing and then used as a bait to screen a random-primed *A. thaliana* meiotic buds cDNA library constructed into pP6 vector. Cloning vectors pB27 and pP6 were derived from the original pBTM116 [26] and pGADGH [27] plasmids, respectively.

The N-*LexA-EaF82*-C bait was tested in yeast and found neither toxic nor autoactivating by itself. Therefore, it was used for the ULTImate Y2H™ screening. A total of 115 million clones (11 fold the complexity of the library) were screened using a mating approach with YHGX13 (Y187 ade2-101::loxP-kanMX-loxP, mata) and L40ΔGal4 (mata) yeast strains as previously described [28]. A total of 271 His^+^ colonies were selected on a medium lacking tryptophan, leucine, and histidine. The prey fragments of positive clones were amplified by PCR and sequenced at their 5′ and 3′ junctions. The obtained sequences were subjected to search the corresponding interacting protein in the GenBank database (NCBI) using a fully automated procedure. A confidence PBS score, which relies on both local and global scores, was assigned to each interaction as described by Formstecher et al. [29]. Briefly, the local score was analyzed by considering the redundancy and independency of prey fragments, as well as the distribution of reading frames and stop codons in overlapping fragments. Then, the global score was analyzed by taking into consideration of the interactions that were found in all the screens performed at Hybrigenics using the same library. The global score indicates the probability of a nonspecific interaction. The PBS scores were divided into six categories (A to E) for practical use purpose. Category A represents the highest confidence, while D stands for the lowest confidence. Moreover, category E specifically flags interactions involving highly connected prey domains previously found several times in screens performed on libraries derived from the same organism, while F stands for a false positive with several of these highly connected domains that have been confirmed as false positives of the technique. The PBS scores have been reported to correlate with the biological significance of interactions well [30,31]. Interacting proteins with PBS scores A, B, C, and D have all been confirmed to have biological relevance independently [29]. Therefore, all interacting proteins with PBS scores A, B, C, and D could be considered as good candidates. 

Using the PBS score system, seven candidates were selected from the obtained EaF82 interacting proteins with score A, B, C, or D (marked as P) for further 1-by-1 Y2H assay to validate the interactions. Seven fragments were extracted from the ULTImate Y2H™ screening and were cloned in-frame with the Gal4 activation domain (AD) into plasmid pP7. The AD construct was checked by sequencing the 5′ and 3′ ends of the inserts. Hybrigenics’ references for these seven preys are listed in Appendix A.

To perform 1-by-1 pairwise Y2H interaction assays, the bait and prey constructs were transformed into the yeast haploid cells L40ΔGal4 (mata) and YHGX13 (Y187 ade2-101::loxP-kanMX-loxP, matα), respectively. These assays were based on the HIS3 reporter gene (growth assay without histidine). As negative controls, the bait plasmid was tested in the presence of empty prey vector (pP7) and all prey plasmids were tested with the empty bait vector (pB27). The interaction between SMAD and SMURF was used as the positive control [32]. Interaction pairs were tested in duplicate as two independent clones (clone 1 and clone 2) for the growth assay. For each interaction, undiluted and 10^−1^, 10^−2^, 10^−3^ dilutions of the diploid yeast cells (culture normalized at 5 × 10^7^ cells/mL) expressing both bait and prey constructs were spotted on several selective media. The DO-2 selective medium, lacking tryptophan and leucine, was used as a growth control to verify the presence of both the bait and prey plasmids. The different dilutions were also spotted on a selective medium without tryptophan, leucine and histidine (DO-3). Four different concentrations (1, 5, 10, and 50 mM) of 3-AT, an inhibitor of the HIS3 gene product, were added to the DO-3 plates to increase stringency and reduce possible autoactivation by the bait and prey constructs. The “DomSight” (Hybrigenics Services, SAS), which displays the comparison of the bait fragment and the Selected Interacting Domain (SID) of the prey proteins with the functional and structural domains (databases of protein domains: PFAM, SMART, TMHMM, SignalP, Coil algorithms) on these proteins, was used for data visualization.

### 2.11. Co-Immunoprecipitation (Co-IP) Analysis

To validate the interaction between EaF82 and AKIN10, co-IP analysis was performed. Each 333 mg ground floral tissues from Arabidopsis vector control or TC transgenic line were resuspended in 1 mL of extraction buffer (50 mM Tris-HCl, pH 7.5 containing 150 mM NaCl, 0.05% Triton-X100, 10% glycerol, and both protease and phosphatase inhibitor cocktail) and kept on ice for 30 min. The samples were then homogenized using a BeadBug 6 microtube homogenizer (Benchmark Scientific) and centrifuged at 15,000× *g* for 10 min. The clear extracts were transferred to new tubes and centrifuged again to remove particulates. The clear extracts were transferred to new tubes.

To prepare the input samples, 100 μL of each clear extract was mixed with 37 μL of 4X LDS sample buffer (NuPAGE™, Invitrogen) and 15 μL of 10X reducing agent. The mixture was heated at 70 °C for 10 min and stored at −20 °C until analysis. The remaining extracts were used for co-IP. First, both control and TC extracts were split into two 250 μL fractions. They were then spiked with or without 20 μg of EaF82-S peptide and divided into two tubes. After that, 10 μg of anti-AKIN10 antibody (Cedarlane, Burlington, NC, USA) was added to one tube, while the other tube received none (as a negative control). Another control was prepared by adding EaF82-S peptide alone to protein A+G magnetic beads. All samples were gently shaken overnight at 4 °C for binding. After that, 50 μL of protein A/G magnetic beads pre-equilibrated with extraction buffer were added and incubated further for 3 h. After binding, all samples were centrifuged at 800× *g* for 1 min to separate the beads. The supernatant was removed and the beads were washed thrice with 400 μL of extraction buffer, followed by two washes with 0.1M Tris-HCl buffer, pH 7.5. The bound complex was released from beads by adding 150 μL of preheated (70 °C) 4X LDS sample buffer and centrifuged at 10,000× *g* for 5 min. The eluates were then transferred to new tubes, mixed with 15 μL of 10X reducing agent (NuPAGE™, Invitrogen), and heated at 70 °C for 10 min. Immunoblotting was then performed to detect EaF82 and AKIN10 in the eluate. Briefly, 20 μL of input and 30 μL of co-IP samples were loaded onto a 12% NuPAGE™ Bis-Tris gel (Invitrogen) and separated at 200 V for 40 min. Following separation, the proteins were transferred onto a PVDF membrane overnight at 4 °C. The membrane was then blocked with fat-free milk powder in PBST (1%) for 1 h at room temperature and incubated with rabbit anti-AKIN10 antibody (1:500 dilution) followed by incubation with Veriblot IP detection reagent (Abcam, 1:500 in blocking solution). Protein bands were detected with Supersignal^TM^ west pico chemiluminescent substrate (Thermo Fisher Scientific, Waltham, MA, USA) and images were captured with the iBright™ CL1500 imaging system (Thermo Fisher Scientific). To detect EaF82, the above blot was stripped with Restore™ stripping buffer (Thermo Fisher Scientific), washed thrice with 1X PBST for 10 min each, and then blocked as described above. The remainder of the procedure to detect EaF82 was the same as described earlier.

### 2.12. Peptide Sequence Analysis and Phylogenetic Tree Construction

The SignalP 5.0 (http://www.cbs.dtu.dk/services/SignalP/ accessed on 5 September 2019) [33] was used to predict the signal peptide on 5 September 2019. The Swiss Institute of Bioinformatics Expasy server (https://web.expasy.org/compute_pi/ accessed on 5 September 2019) [34] was used to predict pI and molecular weight. The construction of two phylogenetic trees was carried out following the information and procedures provided in Campbell and Turner [5] except skipping a manual optimization; and MEGA X [35] was used to perform evolutionary analysis of Arabidopsis clade IV-C RALFs with EaF82.

### 2.13. Statistical Analysis

Statistical analysis was performed with one-way ANOVA and Fisher’s least significant difference (LSD) test. The GraphPad Prism 7 (GraphPad Software) was used for calculating Michaelis–Menten constant (Km) and Vmax as well as statistical best fit value of R square and standard deviation of estimation (Sy.x).

## 3. Results

### 3.1. EaF82 Is a Clade IV RALF-like Peptide

Our previous search for EaF82 homologs using protein BLAST only found an antimicrobial peptide, MiAMP1, as the closest match with E-value of 0.094 and 48% similarity [16]. In the current study, we further analyzed its sequence features and used a recent comprehensive peptide classification system based on peptide structural features and biological functions [8] to find its closely related peptide homologs. In the deduced 120-amino acid EaF82 peptide sequence from the cloned cDNA, a 30 amino acid signal peptide (Figure 1a) was predicted by SignalP 5.0 [33]. EaF82 contains four cysteines at positions 42, 54, 81, and 95 with potential to form two intramolecular disulfide bridges (Figure 1a). Its primary structural features were found to be most similar to a group of CRPs classified as “nonfunctional precursor” by Tavormina et al. [8]. The number of cysteines and the amino acid patterns around disulfide bonds are known to be conserved for proteins having similar folding and function and can be used as the basis for protein classification [36]. Thus, we used these criteria for further classification and revealed that the features of EaF82 were closest to those of RALFs among the listed CRPs, having two predicted intramolecular disulfide bridges and an *N*-terminal signal peptide [37].

The hallmark of RALFs is their ability to rapidly alkalinize tobacco cell culture media upon the addition of an exogenous peptide [7]. To investigate whether EaF82 also possesses a RALF-like alkalinizing activity, EaF82 peptide without a signal peptide was synthesized (designated as EaF82-S) and its activity was measured, including kinetic parameters *Km* and Vmax. EaF82-S exhibited alkalinizing activity with a Vmax of delta pH ~0.4, which was about half of reported values [7], and *Km* of 1 nM (Figure 1b) close to reported values for RALFs [7,10]. The increase in pH peaked at 30 min and returned to baseline after 60 min (Figure 1c). In contrast, the EaF82 alkalinizing activity was abolished when the peptide was reduced and alkylated to break disulfide bonds (a negative control) (Appendix A). The results indicate that EaF82 has RALF-like alkalinizing activity.

To examine the relationship of EaF82 with RALFs, we performed a phylogenetic analysis using the method and sequence information described by Campbell and Turner [5]. We found that EaF82 belonged to clade IV-C (Figure 1d). As Arabidopsis was to be used to heterologously express *EaF82* for functional studies, we further used 13 out of 14 Arabidopsis clade IV-C RALFs from a previous publication [5] to align with EaF82 by MUSCLE [38] and analyzed their relationships using MEGA X [35]. The previously reported AtRALF17 (AT2G32890) [5] was excluded as it lacks common features of RALF family members and is very likely not a RALF peptide [39]. Phylogenetic analysis revealed that EaF82 was closely related to AtRALF8/9/15 (Figure 1e), which are known to be abundantly expressed in pollen [13].

### 3.2. EaF82 Is Expressed and Accumulated in Anthers but Not in Pollen

To investigate its physiological function, a reporter gene *sGFP*(S65T) in frame with *EaF82* driven by an *EaF82* promoter (*EaF82p::EaF82-sGFP* designated as TA) or by a constitutive *CaMV 35S* promoter (*35Sp::EaF82-sGFP* designated as TB) (Appendix A) was transformed into Arabidopsis. The former was used to determine the expression sites of *EaF82* while both of them were used to investigate the effects of EaF82 peptide on Arabidopsis growth and development, with special attention given to pollen development. When the transcriptional and translational expression levels of *EaF82* in TA transgenic seedlings were analyzed by RT-PCR and immunoblotting, the expected sizes of PCR and protein products were detected (Appendix A). These validated TA transgenic seedlings showed GFP signals in roots, especially at the basal and apical meristems of primary and lateral roots (Appendix A), where auxin is known to be accumulated [40]. These results are consistent with our previous finding in which GUS was found to accumulate at the same sites when driven by an *EaF82* promoter [16]. GFP signals were observed in anthers and filaments of mature flowers (opened), but only occasionally in released pollen grains (Figure 2a,b). We also checked the GUS activity in our previously created *EaF82p::GUS* plants [16]. Similarly, less GUS activity was observed in mature flowers than in early stages of closed flower buds (Figure 2c), where high auxin accumulation has been reported [41].

EaF82 is closely related to AtRALF8/9/15 (Figure 1e), which are reported to be abundant in pollen [13]. However, the expression of *sGFP* driven by an *EaF82* promoter was rarely detected in pollen grains (Figure 2b). Thus, its protein level in pollen was further examined in detail. Seedlings with confirmed EaF82 accumulation from four independent TA lines (Appendix A) were transferred to soil and grew into mature plants to collect pollen grains. These TA seedlings grew normally on MS medium (Appendix A) and became mature plants on soil, implying that EaF82 might have no effect on vegetative growth, as reported for AtRALF1/8/23 [6]. Although both EaF82 and GFP proteins were easily detected in flower tissues (Appendix A), collected pollen grains had little detectable EaF82 and GFP proteins, as revealed by immunoblotting (Figure 2d). Consistent with this observation, under the confocal microscope only a few pollen grains displayed GFP signals (Figure 2e). These results indicate that EaF82 was not abundant in pollen. Likely, the observed GFP signals in anthers (Figure 2a) could come from the surrounding tissues instead of pollen grains.

### 3.3. Overexpressing EaF82 Affected Pollen Development and Seed Setting

Interestingly, the above TA lines showed many unpollinated pistils and short siliques on the primary inflorescence stalks (Figure 2f). This observation raised a question of whether EaF82 plays a role in seed setting, since a group of RALFs has been shown to affect the rate of seed setting by inhibiting pollen tube elongation [10]. We collected pollen even though fewer pollen grains are produced in some flowers, and then examined pollen viability and pollen tube elongation. There were no differences in the general features of harvested pollen grains (Figure 2g) as well as in pollen germination ability or in vitro pollen tube elongation (Figure 2h), compared to those of wild type (WT). Therefore, the effects of EaF82 on pollen viability are likely minimal once pollen is produced. These results led us to speculate that those unpollinated pistils and short siliques on the primary inflorescence stalks could be the results of high EaF82 accumulation in the surrounding tissues of pollen sacs at the specific stage of development, causing either no pollen or less pollen available for the pollination. Since the TA lines were driven by an auxin-responsive *EaF82* promoter [16], it is possible that the accumulation of EaF82 along the inflorescence stalks was uneven due to the uneven auxin distribution affected by the growth conditions.

The above speculation that high EaF82 accumulation in the surrounding tissues of pollen sacs might affect pollen development and production was supported by the observations from TB lines with the *EaF82* driven by a strong constitutive *CaMV 35S* promoter. First, although we used the same floral dipping procedure to create TA lines, our transformation efforts with the TB cassette resulted in only two independent lines (TB-1 and TB-2) despite several attempts. The difficulty of TB transformation seemingly implies that the presence of high EaF82 might cause developmental problems in transformed pollen, subsequently affecting the production of transformed seeds. When the two TB lines were grown on MS medium, their seedlings were as normal as the WT (Appendix A). Unfortunately, only TB-2 line showed detection of EaF82 and GFP by immunoblotting (Appendix A). TB-2 plants bore extremely few pollen grains and produced only 1-2 small siliques per plant, even though they produced many flowers (Figure 3a,b). The TB-2 line exhibited more severe seed abortions compared to TA lines (Figure 2f). In order to obtain ~100 siliques, we planted 72 independent plants per subline of TB-2a and TB-2b. The average numbers of seeds per silique in TB-2 plants were reduced to only four, compared to forty-nine in the case of WT (Figure 3c).

Since the major difference between TA and TB lines was the promoters, we further tested promoter activities using multiple independent tobacco (*Nicotiana tabacum*) transgenic lines carrying genetic cassettes *35Sp::GUS* or *EaF82p::GUS* (Appendix A) to compare their *GUS* transcript levels. RT-qPCR results showed ~3.5-fold higher *GUS* expressions in transgenic *35Sp::GUS* plants than in those of transgenic *EaF82p::GUS* (Appendix A), indicating that the activity of *CaMV 35S* promoter is stronger than that of *EaF82* promoter. Thus, the severe seed abortion phenotype in TB-2, driven by the *CaMV 35S* promoter, could probably result from a constitutively higher expression of EaF82-sGFP compared to TA lines driven by the *EaF82* promoter, supporting that the high expression levels of EaF82 may be responsible for the pollenless and reduction in seed yield. 

### 3.4. Overexpressing EaF82 Delayed Tapetum Degeneration during Pollen Development

The observed pollenless phenotype in TB-2 (Figure 3a) suggested that the pollen development was compromised. Considering the observed accumulation of EaF82 in anthers, as well as closed flower buds (Figure 2a–c), where pollen is under development, we first investigated at which stage pollen development was impaired by histological analysis. In Arabidopsis, flower buds are clustered, and the flower development can be divided into 12 developmental stages using a series of landmark events as described by Smyth et al. [42]. Anther development can be further divided into 14 stages based on the visualization of distinctive cellular events under the microscope [43]. According to the key events of each stage, the pollenless TB-2 line was examined, and it was found that stages of microspore mother cells undergoing meiosis and generating tetrads of haploid microspores and microspores releasing from the tetrads (up to stage S8) were similar to those in WT (Figure 3d). The abnormality was noticed at stages S11 and S12. At S11, the complete disappearance of tapetum occurred in WT but not in TB-2, and hence most of the TB-2 pollen grains were aborted at S12 (Figure 3d). These observations were further confirmed in another TB-2 flower cluster (Appendix A), indicating that EaF82 impairs tapetum degeneration and pollen development. 

### 3.5. EaF82 also Affected Pollen Development and Seed-Setting in Tobacco Plants

To verify that the observed effects of EaF82 on pollen development and seed setting were not limited to Arabidopsis, a genetic cassette *35Sp::EaF82* (designated as TC) (Appendix A) was expressed in *N. benthamiana* plants. Because the tobacco leaf disc transformation method is efficient, this approach was used to generate 15 independent transgenic lines (T1–T15), with seven of them exhibiting a 3 to 1 segregation. Out of these seven lines, four (T2, T3, T8, and T11) had detectable EaF82 in their leaves (Figure 4a) and flowers (Figure 4c) as revealed by immunoblotting. Mature anthers of these four transgenic lines were often shriveled in appearance with no or fewer pollen grains, while those of the vector control lines were dehisced with many released pollen grains (Figure 4b). Transgenic flowers were either not fertilized to develop seed pods or partially fertilized to develop small seed pods compared to those normal pods produced from the control lines (Figure 4d). Among produced pods, the seed numbers per pod of lines T2, T3, T8, and T11 were reduced by 42%, 22%, 50%, and 58%, respectively, compared to the control lines (Figure 4e). The results from these transgenic lines indicate that EaF82 also inhibits pollen development and seed setting in tobacco plants. 

### 3.6. Overexpressing EaF82 Induced Transcriptome Changes

To elucidate the observed delayed tapetum degeneration during pollen development, RNAseq analysis was performed to gain molecular insights of EaF82-associated transcriptional alteration. To ensure pollen abortion is associated with overexpressed EaF82 but not sGFP, Arabidopsis transgenic lines with the TC genetic cassette (*35Sp::EaF82*) without *sGFP* (Appendix A) were created. Similar to TB, in which the high expression of EaF82 resulted in nearly no transformed seeds, transformation with TC genetic cassette was also challenging and produced only two lines (TC-1 and TC-2) with detectable EaF82 levels (Figure 5a,b). They grew normally during the vegetative stage compared to WT (Figure 5c) but exhibited seed abortion with most undeveloped siliques (Figure 5d), such as the TB line. Their unopened flowers had detectable EaF82 (Figure 5e) and later could develop a few viable pollen grains. When a few pollen grains were harvested for germinating, the pollen tube growth was normal (Figure 5f,g). Among developed TC siliques, however, many of them were short in length compared to most WT siliques (Figure 5h). A large-scale measurement using siliques from 10 primary stems per line showed that approximately 20.5% of TC-1 and 27.8% of TC-2 siliques did not develop longer than 0.3 cm (Figure 5h), which contained no seeds at all (Appendix A). Only 16.6% of TC-1 and 18.8% of TC-2 siliques were longer than 1.0 cm with some seeds while that of WT was 88.5% (Figure 5h).

To gain molecular insights into transcriptional changes during pollen development, we analyzed the gene expression profiles of the unopened flower bud clusters covering all anther developmental stages from TC-1 and TC-2 lines by RNAseq, along with a vector control line. Two TC lines were used to perform double verification of any observed differentially expressed genes (DEGs). The numbers of reads were between 18.3 to 23.0 million (Appendix A), and listed genes reported as FPKM (fragments per kilobase million) were 28,296 (Appendix A), covering ~90% of total nuclear, mitochondria, and chloroplast genes (Appendix A). The numbers of common DEGs in TC-1 and TC-2 either increased or decreased 2 fold with adjusted *p*-value for false discovery rate (FDR) < 0.05 compared to the vector control were 158 and 1197, respectively (Figure 5i; Appendix A). Hierarchical cluster analysis of 74 downregulated DEGs with log_2_FC ≤ −1.5 (reduced ≥ 2.8 fold) plotted in a heatmap with R function showed a strong correlation between TC-1 and TC-2 lines (Appendix A). These results demonstrated that overexpressing *EaF82* induced ~5% nuclear gene expression changes ≥ 2 fold in the early development of flowers. 

### 3.7. Overexpressing EaF82 Suppressed the Expression of Genes Involved in Cell Wall Modifications and pH Changes

To determine the affected pathways and related genes in overexpressing *EaF82* lines, the identified 158 upregulated and 1197 downregulated DEGs were subjected to Gene Ontology (GO) analysis using PANTHER v16 [21], and some selected DEGs were validated using RT-qPCR. In the upregulated DEGs, only regulation of the developmental process (GO:0050793) in the biological process and sequence-specific DNA binding (GO:0043565) in molecular function were significantly enriched with a similar group of genes (Appendix A), including flowering-related genes *AGL19, TFL1, AGL20 (SOC1), AGL24, AGL42, FD*, and *SAP* (Appendix A) [44,45]. All four upregulated AGAMOUS-LIKE DEGs *AGL19, AGL20, AGL24*, and *AGL42* as well as 27 downregulated DEGs involved in four different categories (Table 1 and Table 2) were selected for validation by RT-qPCR. Their transcriptional changes from RNAseq analysis were well confirmed by the RT-qPCR results with similar fold decreases (Table 1 and Table 2).

In the downregulated DEGs, notably enriched GO terms related to the observed characteristics of transgenic plants were those involved in the regulation of pH and cell wall modification (Figure 5j; Appendix A). The overrepresented DEGs include those encoding 29 pectin methylesterases (PMEs)/pectin methylesterase inhibitors (PMEIs), along with five H^+^-ATPases and sixteen cation/H^+^ antiporters (Table 1; Appendix A), which are known to play important roles in regulating pH changes and modulating the cell walls as an adaptation to stresses during plant development [46,47,48,49]. Among the downregulated pollen specific *PMEs* (Appendix A), *PPME1* and *PME48* are regulated by RGA, a GA repressor DELLA [50]; while *PME5/VGD1*, *PME4/VGDH1*, and *VGDH2* are regulated by MYB80, a transcription factor regulating both tapetum and pollen development [51]. These downregulated DEGs, together with reported roles of RGA and MYB80 at the late stage of tapetum degeneration and pollen development [50,51], suggest that cell wall modification was altered during the pollen development, supporting observed impaired tapetum degeneration and pollen abortion (Figure 3d). Moreover, seven DEGs (*AGP*5/6/11/14/23/24/40) encoding highly glycosylated arabinogalactan proteins (AGPs), which play key roles in pollen wall formation [52,53], were also downregulated (Appendix A). These downregulated *AGP*s present a strong correlation with the observed aborted pollen at stages S11 and S12 (Figure 3d; Appendix A).

### 3.8. EaF82 Suppressed the Expression of Genes Involved in Tapetum Degeneration and a Group of AtRALFs

Pollen develops inside the anther locules [54]. The innermost layer of anther locules is the tapetum, which is the main tissue providing nutrition and enzymes for pollen development and pollen wall formation [55,56]. There is mounting evidence suggesting that the pollen developmental process is closely linked to the development of tapetum [56,57], with the latter being divided into three developmental stages: tapetum differentiation, tapetum formation, and tapetum degeneration through program cell death (PCD) [55]. Tapetal cells appear at stage S5 and their degradation is initiated at the stage S10 [43]. Our histological results of pollen development (Figure 3d; Appendix A) showed that overexpressing EaF82 interrupted tapetum degeneration, but not tapetum differentiation and formation. This was supported by RNAseq data, which showed that many known genes involved in early stages of tapetum differentiation and tapetum formation [56] were detected but not differentially expressed (Appendix A), whereas the expression of *CEP1*, involved in late stage of tapetum degeneration, was found to be reduced ~16 fold (Appendix A). *CEP1* encodes a papain-like cysteine protease, which participates in tapetal cell wall hydrolysis, leading to the tapetal cell wall degeneration [58]. Therefore, ~16-fold downregulated *CEP1*, together with a large number of cell wall modification associated genes (Appendix A), supports the observed impairment of tapetal cell degradation (Figure 3d; Appendix A).

In addition to the above genes, seven *AtRALFs,* including *AtRALF*4/8/9/19/25 observed to be downregulated in four tapetum mutants [12], were also found in our down DEGs (Table 2; Appendix A). Their downregulation was further confirmed by RT-qPCR showing similar fold reductions in their expressions (Table 2).

### 3.9. Overexpressing EaF82 Decreased Proteasome Activity and ATP Levels

The tapetum degeneration is a process that resembles apoptosis-like PCD [57] and is essential for proper pollen development [59,60]. Both proteases and proteasomes are critical to the progression of tapetum PCD [59,61]. In addition to *CEP1*, seven additional downregulated DEGs encoding proteases (Appendix A) suggest that PCD processes may be defective. As we did not find any proteasome genes in our DEGs, we examined the translational abundance of three subunits (Rpn6, Rpn10, and Rpt5) of the proteasome with immunoblotting and proteasome activity in the early development of flowers. Among them, Rpt5a is one of the six AAA-ATPases of proteasome and essential for pollen development [62]. Immunoblotting showed no differences in their protein abundances (Figure 5k). However, the proteasome activity in TC-1 and TC-2 lines was reduced by 33% and 49% compared to the WT, or by 48% and 60% compared to the vector control, respectively (Figure 5l). 

The results of reduced proteasome activity prompted us to examine ATP levels further because both the PCD process and proteasome assembly and activities are ATP-dependent [63,64,65]. Moreover, ATP serves not only as an intracellular energy molecule but also as a signal molecule in the extracellular matrix of plant cells through coordinating with Ca^2+^ and ROS [66,67]. ATP defective mutants are male sterile [68,69], while elevated ATP increases seed yields [70]. Using the same tissues used for examining proteasome subunits and proteasome activity, we found that ATP levels were indeed reduced by 38% and 57% compared to the WT, or by 46% and 63% compared to the vector control, respectively (Figure 5m). Their similar trends of reduction suggested that the reduced proteasome activity was likely the result of low ATP supply at the early developmental stages of flowers, even though no DEGs directly linked to ATP production (such as genes coding for ATP synthases) were found. Consistent with the ATP deficiency in TC lines, many downregulated DEGs, which encode proteins directly or indirectly affected by ATP, such as six ATP-binding cassette (ABC) transporters and five H^+^-ATPases, as well as a group of ATP-binding receptor-like protein kinases (Table 1; Appendix A), were observed. All these results together suggest that overexpressing EaF82 lowered ATP levels and triggered the downregulation of genes encoding for ATP-binding proteins, which in turn affected PCD activity and resulted in delaying tapetum degeneration. 

Additionally, tapetum degeneration is essential for releasing nutrients to support pollen grain development and maturation [55,56]. In the DEG, a large number of genes encoding transporters and transmembrane proteins for shuttling sugars, amino acids, and ions were also found to be downregulated (Appendix A), suggesting that the nutrient transport was probably limited to the developing pollen grains. 

### 3.10. AKIN10 Is an Interacting Partner of EaF82

To understand how overexpressing EaF82 causes a decrease in ATP levels and the downregulation of so many genes, we conducted Y2H screening followed by a one-by-one interaction validation assay with *EaF82-S* as the “bait” and a cDNA library made from Arabidopsis mitotic flower buds as the “prey” to identify EaF82 interacting partners. Using the PBS (predicted biological score) system, a total of 46 EaF82 interacting proteins with a score of A, B, C, or D were obtained as good candidates (Appendix A). Based on their biological functions and cellular components from published studies, seven candidates, namely ABCF4, ALATS, FKBP-like peptidyl-prolyl cis-trans isomerase family protein, PAPP2C, TCH4, AKIN10, and SYTA (Appendix A), were selected to perform a one-by-one Y2H validation assay. The assay results were shown in Appendix A and summarized in Appendix A. Among the seven candidates, we found three—ABCF4, PAPP2C, and AKIN10—to have the strongest interactions. ABCF4 (AT3G54540, also named as AtGCN4) is an ATPase and regulates plasma membrane H^+^-ATPase activity [71]; PAPP2C (AT1G22280) is a protein phosphatase [72]; while, AKIN10 (AT3G01090) is a major cellular energy sensor in plants and orthologous to mammalian AMP-activated protein kinase (AMPK) [73].

Given the key role of AKIN10 as the energy sensor in plants and its orthologous to mammalian AMPK [73], the interaction between AKIN10 and EaF82 (Figure 6a) could be critical and is likely to be responsible for observed ATP deficiency and aberrant tapetum degradation. ABCF4 could be also involved in the EaF82-induced intracellular pH increase leading to cell wall modification. To elucidate the observed induced ATP deficiency (Figure 5l), we focused on AKIN10 and performed co-IP analysis to validate its interaction with EaF82. 

An initial attempt to directly detect EaF82 in the pull-downed AKIN10 and EaF82 complex from extracts of TC transgenic floral tissues was unsuccessful. This could be due to low levels of such a complex present in the TC extracts with only limited EaF82 interacting with AKIN10. Since the EaF82-S without signal peptide was used for alkalinizing activity assay and in Y2H analysis, the EaF82-S peptide was spiked into both the vector control (C) and TC transgenic line extracts to perform the co-IP assay. The idea was that if AKIN10 is indeed an interacting partner of EaF82, the added EaF82-S peptide should form a complex with endogenous AKIN10 in both lines, and with co-IP with anti-AKIN10 antibody, the complex should be able to pull down. Consistent with this idea, we were able to pull down the AKIN10-EaF82-S complex from both the vector control and TC transgenic line extracts (Figure 6b; lanes 4 and 7). Under the same incubation conditions without the anti-AKIN10 antibody, neither AKIN10 nor EaF82 was detected (Figure 6b; lanes 3 and 6), while EaF82 was not detected when EaF82-S was incubating with protein A/G magnetic bead suspension (Figure 6b; lane 8), indicating that no adsorption of AKIN10 or EaF82-S to protein A/G magnetic beads occurred. The detected EaF82 (Figure 6b; lane 4 and 7) could only be pulled down by the anti-AKIN10 antibody bound protein A/G magnetic beads when it was present in the complex with AKIN10. These results confirm that AKIN10 can interact with EaF82, supporting the Y2H finding. 

### 3.11. Elevated Levels of AKIN10 in EaF82 Transgenic Flowers with ATP Deficiency

In mammals, the level of AMPK increases when the AMP/ATP ratio is high, resulting in phosphorylation of multiple downstream targets to increase ATP production and decrease ATP consumption [74]. In plants, the level of AKIN10 (AMPK homolog) was found to increase as the AMP/ATP ratio increased, while its transcriptional levels remained constant [75]. In addition, it has been reported that the overexpression of *AKIN10* resulted in late flowering and defective silique development [76,77]. For the identified EaF82 interacting partner AKIN10 (Figure 6a, b), its gene *AKIN10* and another member, *AKIN11*, were detected in our RNAseq analysis but were not differentially expressed (Appendix A). AKIN10 has also been reported to be degraded in a proteasome-dependent manner [78] and induced with low energy conditions such as dark growth and hypoxia [75,79]. To determine whether reduced ATP levels and proteasome activity cause an elevation in AKIN10 levels, we used immunoblotting with anti-AKIN10 antibody and discovered that TC-1 and TC-2 had an average of ~4-fold higher levels of AKIN10 and ~2-fold higher levels of AKIN11 compared to WT and vector control (Figure 6c). These results are in agreement with the low ATP and reduced proteasome activity in the transgenic lines (Figure 5l,m).

## 4. Discussion

*EaF82* is a novel gene isolated from variegated ‘Golden Pothos’ plants with no known function [16]. It contains no intron and encodes a 120 amino acid long peptide [16] with a 30 amino acid signal peptide at the *N*-terminus and four cysteine residues (Figure 1a), sharing the typical features with numerous plant RALFs [37]. In the present study, EaF82’s identity as a member of the RALF family was verified by its rapid alkalinization capacity (Figure 1b,c), a hallmark of most RALFs [7], and its overexpression affecting cellular pH regulation and cell wall modification related genes (Table 1; Appendix A). Phylogenetic analysis indicated that it belongs to clade IV (Figure 1d), the least characterized group of RALFs. Clade IV members are mainly expressed in reproductive tissues [5], but under certain conditions, they may be induced in other tissues. For example, AtRALF8 was reported to be abundant in pollen [13] while it was also found to be induced in roots during drought and nematode infection [80]. *EaF82* expression was previously only found to correlate with IAA distribution in vegetative parts of ‘Golden Pothos’ [14,16], but was unknown in floral organs/tissues. Here, we demonstrated that the ectopic expression of EaF82 in Arabidopsis delayed tapetum degeneration resulting in low pollen production and seed setting, while appearing not to inhibit growth in vegetative parts (Figure 5c), including seedlings (Appendix A).

In flowering plants, pollen development occurs in the anther through a complex process from initial microsporogenesis to the production of mature pollen grains [43,81]. During this process, the sporophytic anther tissues, particularly the tapetum cell layer, play an essential role in regulating the developmental process and providing materials for pollen wall formation [12,81]. Recently, two tapetum-derived small peptides, Casparian Strip Integrity Factor Three (CIF3) and Four (CIF4), were shown to play an essential role in normal pollen wall deposition and tapetum function [82]. Our results also suggest that EaF82 may play an essential role in tapetum degeneration and pollen development. Histological analysis showed that the tapetum degeneration was delayed by EaF82 in TB lines (Figure 3d; Appendix A). The involvement of EaF82 in tapetum degeneration and pollen development was further supported by our RNAseq and RT-qPCR results, where we found that a group of *RALFs* reported in Arabidopsis tapetum mutants [12] was also downregulated in our transgenic lines. In addition, *CEP1* and seven additional protease genes identified in our downregulated DEGs might be involved in impairing tapetum degeneration and causing a reduction in nutrient supply to timely support pollen wall formation. Moreover, the downregulation of seven *AGPs* important for pollen wall formation [52] and many DEGs involved in nutrient transports (Appendix A) supports the observed pollen formation defect (Figure 3d; Appendix A). In addition, many genes encoding PME and PMEI for cell wall modification [46] and a group of auxin-responsive genes essential for pollen development [83] were in downregulated DEGs (Appendix A). Twenty-one transcription factors, including some known to regulate pollen formation and be pollen-specific, were also in downregulated DEGs (Appendix A). All of these downregulated genes lead us to believe that the RALF-like peptide EaF82 compromised pollen formation via the impairment of tapetum degeneration. Further investigation of cell wall changes in tapetum cells and pollen grains during the pollen formation process in greater detail is needed, which might provide direct evidence for our observed delay in tapetum degeneration.

Furthermore, our RNAseq and RT-qPCR data showed that the overexpression of *EaF82* resulted in the repression of seven *AtRALF* genes (*AtRALF4/8/9/15/19/25* and *At4g14020*) (Table 2). This intriguing phenomenon could be due to a feedback mechanism for regulating the expression of endogenous *AtRALFs*. Once these seven *AtRALF* propeptides undergo proteolytic processing to become mature peptides, they all have calculated molecular weights between 6.6 and 8.7 kD and pIs (isoelectric points) greater than 7 (9.3–10.6) (Table 2), indicating that they are basic small peptides. Mature EaF82 (EaF82-S) also has a calculated pI of 8.7. It is possible that the overexpressed EaF82 peptide with positive charge at higher accumulation levels might compete with these AtRALFs to bind to their receptors and affect regulatory pathways leading to their transcriptional repression. These *AtRALFs* might be responsible for cell-to-cell communication between tapetal cells and pollen cells, an important process for pollen development. Nevertheless, the regulatory mechanism of *EaF82* on the downregulation of these *AtRALFs* and their involvements needs further study.

Concerning the action mode of EaF82, the Y2H assay identified AKIN10 as its interacting partner (Figure 6a). AKIN10 is a catalytic α-subunit of an SnRK1 complex that comprises an *N*-terminal Ser/Thr kinase domain, an adjacently linked ubiquitin-associated (UBA) domain, and a large *C*-terminal regulatory domain involved in the interaction with the regulatory (β and γ) subunits and upstream phosphatases [79]. *AKIN10* is expressed in flowers, anthers, and pollen [84,85]. Under normal conditions, AKIN10 is localized mainly in cytosol [86] where EaF82 is also localized [16], supporting our detected interaction between EaF82 and AKIN10 in yeast (Figure 6a). Our co-IP assay validated the interaction between EaF82 and AKIN10 (Figure 6b). Concerning how EaF82 becomes a mature peptide and interacts with AKIN10, we speculate that EaF82 produced in one cell may be released into the extracellular space, where it then interacts with a receptor localized on the cell surface of another cell to cross the plasma membrane, and possibly its signal peptide is cleaved during or after its translocation across the membrane. After entering the cell cytosols, the mature peptide (as EaF82-S rather than its nascent form) might then interact with kinase AKIN10. Nevertheless, further studies are warranted to gain a better understanding of the mechanism and possible effects of this interaction. Based on obtained results, it is reasonable to assume that in EaF82 overexpressed lines, mature EaF82 binds to AKIN10 and modulates SnRK1 kinase activity and phosphorylation of downstream targets leading to transcriptomic changes. Although we did not examine any phosphorylation changes of proteins downstream of SnRK1, we did observe ~5% transcriptomic changes at the early developmental stages of transgenic flowers overexpressing *EaF82*. Our Y2H results showed that EaF82 interacted with AKIN10 through its *C*-terminal end, which also contains kinase-associated domain one (KA1, position 492-533) (Appendix A). The KA1 domain is a conserved domain of AMPKs from yeast to humans, which is involved in autoinhibition, the tethering of acidic phospholipids, and the binding of peptide ligands [87]. Whether EaF82 binds to the KA1 domain to affect SnRK1 activity needs further investigation, as the function of the KA1 domain in AKIN10 is still unclear [79].

Our previous study indicated a correlation between the differentially expressed *EaF82* and the formation of color-defective tissues in variegated leaves of *E. aureum*. We proposed that the light exposure on developing leaves could drive the auxin flux unevenly, causing differential expression of *EaF82*, whose promoter is responsive to auxin [16]. However, whether the identified function of EaF82 inducing ATP deficiency could play a role in chloroplast biogenesis and contribute to leaf variegation formation is still inconclusive. In the present study, the expression of EaF82 in Arabidopsis and tobacco plants did not cause variegation in the leaves, at least indicating there are some other factors, apart from EaF82, involved in variegated formation. Those factors might be lacking in both Arabidopsis and tobacco. Future studies are warranted to determine whether EaF82 interacts with AKIN10 or other factors to cause leaf variegation formation.

Nevertheless, our study for the first time shows that EaF82 is a RALF-like peptide and plays an important role in tapetum degeneration and pollen development. As mentioned above, Pothos plants are among the most popular houseplants used worldwide for interior decoration. Because they hardly flower in nature, breeding through hybridization has not been conducted. As a result, there have been only six cultivars in commercial production [88]. Recently, we found that the shy flowering characteristic of Pothos is due to the deficiency in gibberellin (GA) production, and foliar application of GA can induce flowering [15], which could lead to Pothos breeding through traditional breeding. However, the present study shows that EaF82 may compromise pollen formation via the impairment of tapetum degeneration. Thus, it appears that the breeding of new cultivars of Pothos may have additional hurdles and further research is needed to overcome these difficulties.

## 5. Conclusions

Our results demonstrate that RALF-like peptide EaF82 delayed tapetum degeneration to impair pollen development through induced ATP deficiency. Its action mode may involve binding to AKIN10, leading to alterations of transcriptome and energy metabolism, and thereby inducing ATP deficiency and impairing pollen development. Our study reveals a new regulatory role for the RALF peptide in tapetum degeneration and pollen development.

## Figures and Tables

**Figure 1 cells-12-01542-f001:**
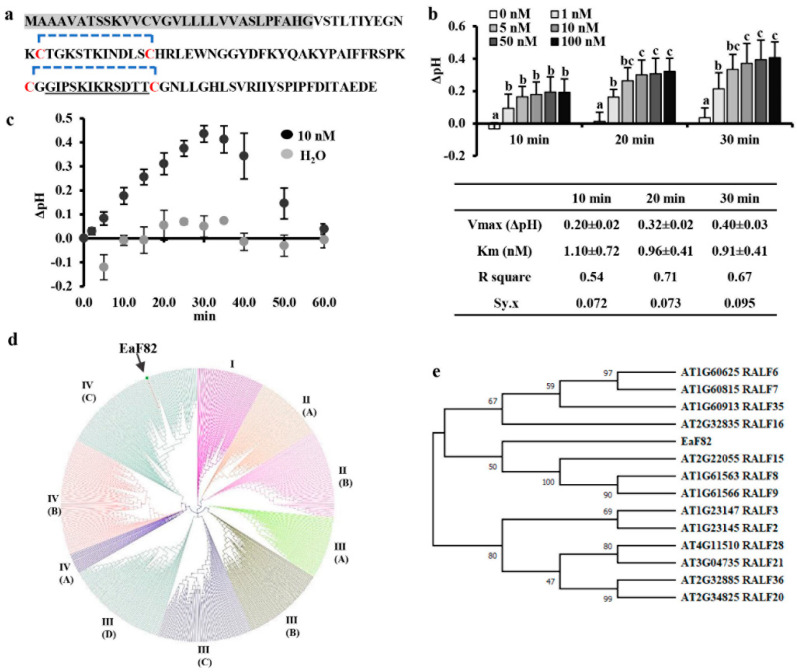
EaF82 peptide and alkalinization assay. (**a**) Amino acid sequence. The predicted signal peptide is highlighted in gray. Four cysteines (C) are marked in red with predicted potential intramolecular disulfide bridges indicated with blue brackets. The sequence used for making an EaF82 antibody [16] is underlined. (**b**) Alkalinization assays of EaF82-S (EaF82 without signal peptide). Six different concentrations (0, 1, 5, 10, 50, and 100 nM) were tested and the pH changes (Δ pH) were measured after 10, 20, and 30 min. Data plotted were the average of five independent experiments ± SD. The Km and Vmax are listed below. Data marked with the same letter are not significantly different by the LSD test at 5% level of significance. (**c**) Alkalinization activity measured for 60 min. Data represent the average of five independent experiments ± SD. (**d**) Phylogenetic tree of EaF82 among 795 RALFs from 51 plant species. Clade I, II, III, and IV as well as their subgroups are categorized following Campbell and Turner [5]. (**e**) Phylogenetic tree of EaF82 with clade IV-C of AtRALFs.

**Figure 2 cells-12-01542-f002:**
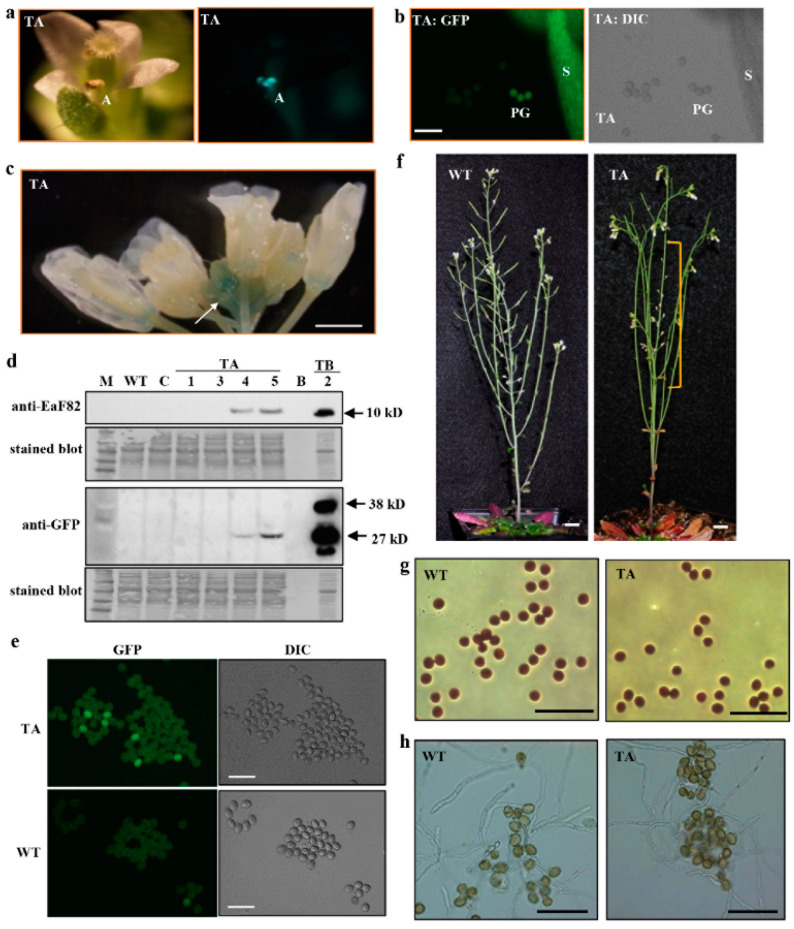
The functional characterization of Arabidopsis transgenic lines carrying *EaF82* promoter driving *EaF82-sGFP* (TA) or *GUS*. (**a**) The newly opened flower (left) shows GFP signal in anthers (right). A: anther. (**b**) Confocal microscopy shows GFP (left) and DIC (differential interference contrast, right) in stamen and some pollen grains. S: stamen; PG: pollen grains; Bar = 100 µm. (**c**) GUS staining of the flowers of *EaF82p::GUS*. White arrow indicates GUS activity at the early developmental stages of flowers. Bar = 1.5 mm. (**d**) Immunoblot of pollen proteins against anti-EaF82 and anti-GFP antibodies. Stained blots show protein loading. TA-1, -3, -4, and -5: four independent lines. TB-2: Proteins isolated from flowers of Arabidopsis transgenic *35Sp::EaF82-sGFP*. B: blank. C: vector control. WT: wild type. M: protein size marker. (**e**) Confocal microscopy detecting GFP in collected pollen grains from Arabidopsis transgenic *EaF82p::EaF82-sGFP* (TA) plants. WT is a negative control. Bar = 100 µm. (**f**) A 10-week-old TA plant (right) bears aborted siliques in primary inflorescence stalk (yellow bracket) compared to normal WT (left). Bar = 1 cm. (**g**) Collected pollen grains from fully opened flowers stained with iodine-potassium iodide. (**h**) Germinated pollen under germination medium. Bar = 100 µm.

**Figure 3 cells-12-01542-f003:**
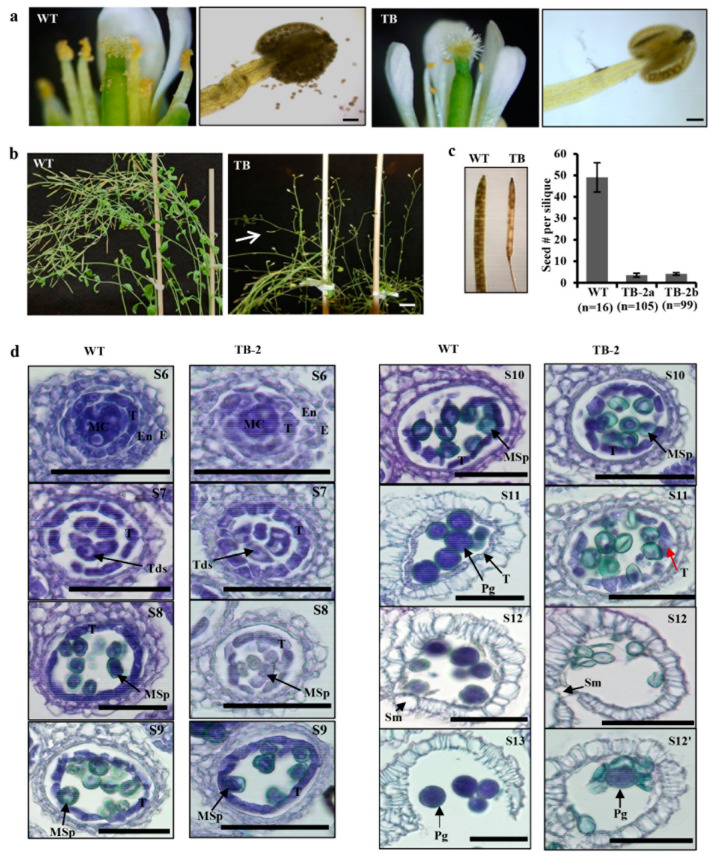
Male gametophyte and pollen development of Arabidopsis transgenic *35Sp::EaF82-sGFP* lines (TB) comparing to wild type (WT). (**a**) Fully opened flower (left) and stained anther with iodine-potassium iodide (right) of WT and TB line, respectively. Bar = 100 µm. (**b**) TB plants (right) produced no siliques with occasionally observed small silique (white arrow) compared to WT (left). Bar = 1 cm. (**c**) TB silique carries fewer seeds than that of WT (left). The numbers of seeds per silique are plotted as the mean ± SD (right). n: numbers of siliques. (**d**) Male gametophyte development of TB line compared to WT. At the stage S11, the undegenerated tapetum (red arrow) was observed in TB line. At the stages S11 and S12, underdeveloped pollen was stained in light green, while mature pollen is in dark blue that occasionally was observed in TB line (S12′). Bar = 50 µm. E: epidermis; En: endothecium; MC: mitotic cell; MSp: microspores; Sm: septum; T: tapetum; Tds: tetrads.

**Figure 4 cells-12-01542-f004:**
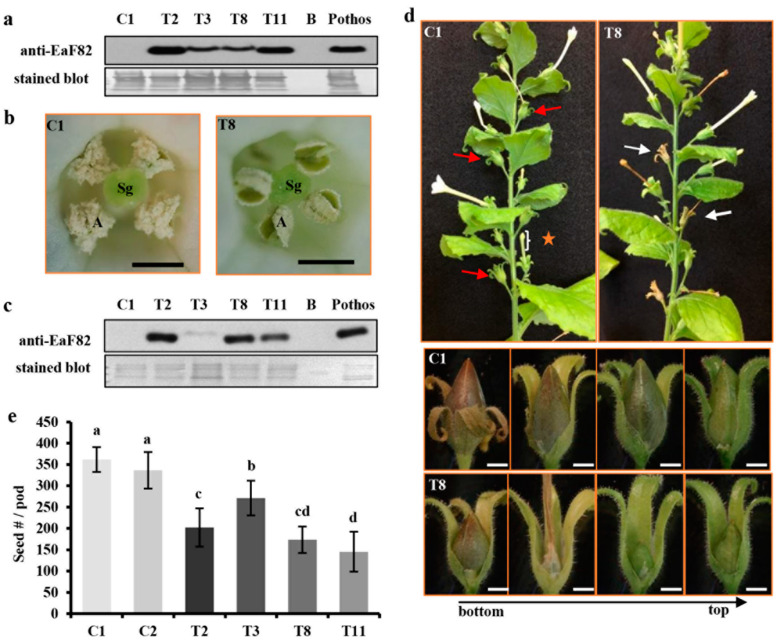
The functional characterization of tobacco transgenic *35Sp::EaF82* lines (T2, 3, 8, and 11) comparing to two independent vector control lines (C1 and C2). (**a**) Immunoblot of leaf proteins against anti-EaF82 antibody. Stained blot shows protein loading. (**b**) T8 (right) and C1 (left) opened flowers. Sg: stigma. Bar = 200 µm. (**c**) Immunoblot of unopened flower proteins against anti-EaF82 antibody. The unopened flowers are as indicated in (**d**) with orange star. (**d**) Transgenic plants with normal (red arrow) and aborted (white arrow) seed pods. Enlarged seed pods are shown below. (**e**) Seed counts per pod. Data plotted are the average from six independent plants per line using 10 seed pods per independent plant ± SD. Data marked with the same letter are not significantly different by the LSD test at 5% level of significance. Bar = 200 µm.

**Figure 5 cells-12-01542-f005:**
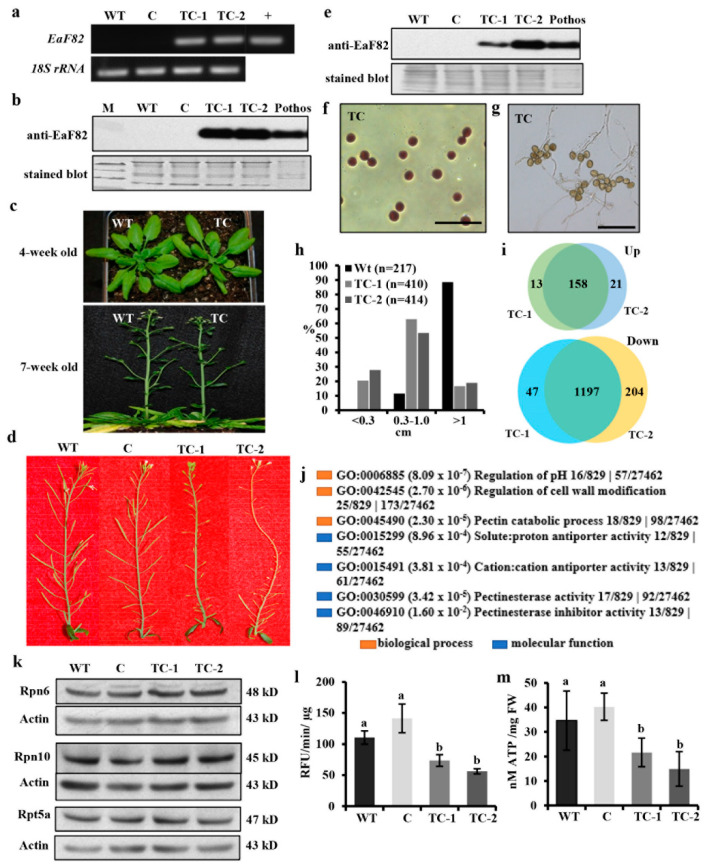
The functional characterization and RNAseq analysis of Arabidopsis transgenic *35Sp::EaF82* lines (TC-1 and -2) compared to wild type (WT) and vector control (C). (**a**) RT-PCR of leaf tissues using primer pair specific to *EaF82* and *18S rRNA*. +: plasmid DNA. (**b**) Immunoblot of leaf proteins against anti-EaF82. Stained blot shows protein loading. M: protein size marker. (**c**) Normal growth of TC and WT plants. (**d**) Aborted siliques in TC-1 and TC-2. (**e**) Immunoblot of proteins from unopened whole flowers against anti-EaF82. (**f**) Stained pollen and (**g**) germinated pollen. The descriptions are the same as in Figure 2. Bar = 100 µm. (**h**) Histogram of the silique lengths. Data plotted are the percentages of total siliques. n: numbers of siliques. (**i**) The number of up and downregulated DEGs (≥2-fold) in TC-1 and -2 lines. (**j**) Enriched GO terms from downregulated DEGs related to pH regulation and cell wall modification. (**k**) Immunoblots of three subunits of the proteasome in the early developmental flowers against anti-Rpn6, anti-Rpn10, and anti-Rpt5a antibodies showed no significant difference among all samples. (**l**) Proteasome activity. Data plotted are the average of three biological replicates ± SD. RFU: relative fluorescence units. (**m**) ATP content. Data plotted are the average of four biological replicates ± SD. FW: fresh weight.

**Figure 6 cells-12-01542-f006:**
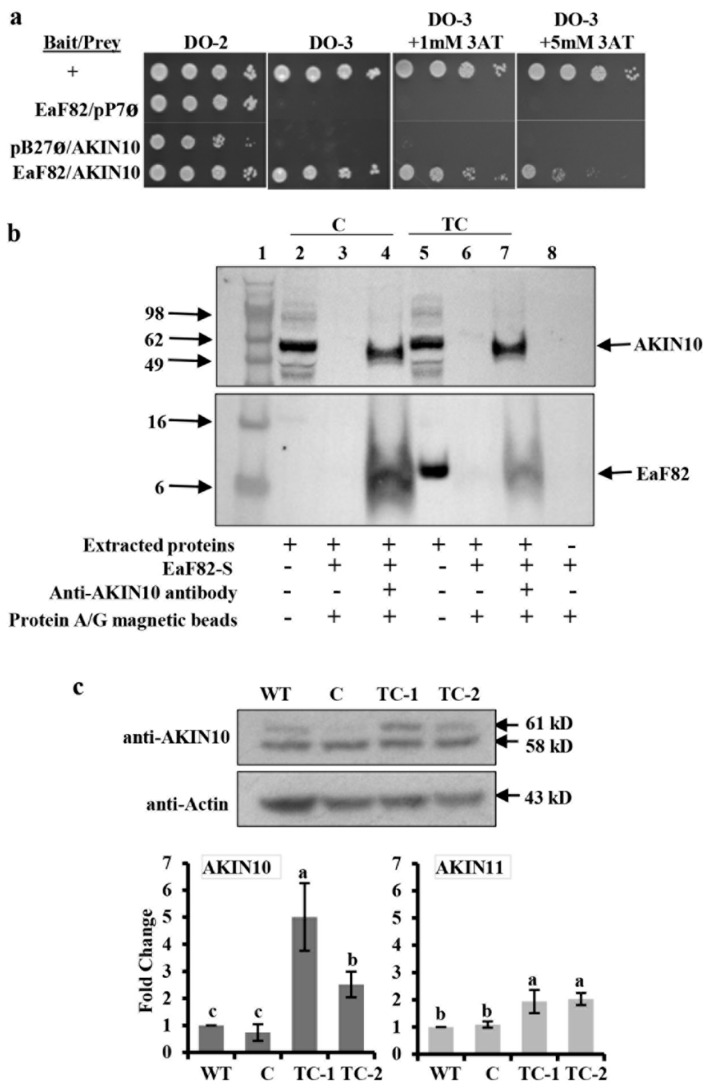
Identification and characterization of an EaF82 interacting partner. (**a**) Yeast growth tests of 1-by-1 Y2H assay on selective medium without (DO-2) or with (DO-3) histidine and 3-aminotriazole (3AT). Supporting information is detailed in Appendix A. +: positive interaction; pB27ø and pP7ø: empty vectors. (**b)** Coimmunoprecipitation (co-IP) analysis to validate the Y2H results of EaF82 and AKIN10 interaction. Protein extracts from floral tissues of vector control (C) and TC transgenic lines were spiked with (+) or without (−) EaF82-S. The complex was pulled down with (+) or without (−) anti-AKIN10 antibody. The immobilized anti-AKIN10 on protein A/G magnetic beads could pull down EaF82 and AKIN10 complex from both C and TC lines (lanes 4 and 7). EaF82 was undetectable without immobilized anti-AKIN10 antibody (lanes 3 and 6). Lane 1: protein size marker. Lanes 2 and 5: extracted proteins from C and TC lines. Lane 8: EaF82-S alone bound to protein A/G magnetic beads. (**c**) A representative immunoblot of proteins from unopened flowers against anti-AKIN10 that detects both AKIN10 (61 kD) and AKIN11 (58 kD). The average of band intensity is plotted as three independent experiments ± SD (below). Data marked with the same letter are not significantly different using the LSD test at 5% level of significance.

**Table 1 cells-12-01542-t001:** RT-qPCR of a subset of selected DEGs.

Gene_ID ^a^	Gene Description ^b^	TC-1 (n = 3)		TC-2 (n = 3)	
		RNAseq (FC)	RT-qPCR (FC ± SD)	RNAseq (FC)	RT-qPCR (FC ± SD)
** *Flowering and pollen development related* **					
AT2G45660	*SOC1/AGL20* (AGAMOUS-like)	2.52	2.15 ± 0.27	2.55	2.57 ± 0.13
AT4G24540	*AGL24* (AGAMOUS-like)	3.02	2.96 ± 0.35	3.27	3.80 ± 0.86
AT5G62165	*AGL42* (AGAMOUS-like)	2.10	2.13 ± 0.05	2.10	2.35 ± 0.18
AT4G35900	*FD* (Basic-leucine zipper transcription factor)	3.26	1.80 ± 0.16	3.84	2.24 ± 0.07
AT1G19890	*MGH3* (male-gamete-specific histone H3)	−2.93	−4.34 ± 1.32	−4.91	−4.34 ± 1.65
AT1G19960	transcription factor	−5.10	−6.60 ± 0.87	−7.50	−8.55 ± 2.86
AT1G21000	PLATZ family transcription factor	−4.16	−4.13 ± 1.52	−4.42	−3.86 ± 1.08
AT1G35490	bZIP family transcription factor	−5.35	−7.04 ± 3.20	−7.11	−7.31 ± 2.50
AT2G36080	*ABS2/NGAL1* (AP2/B3-like transcription factor)	−4.40	−5.75 ± 1.67	−4.58	−5.69 ± 1.69
AT1G24520	*BCP1* (*Brassica campestris* homolog pollen protein 1)	−6.65	−3.15 ± 1.65	−14.91	−98.37 ± 5.98
AT5G17480	*PC1/APC1/CML29* (pollen calcium-binding protein 1)	−2.44	−4.17 ± 1.85	−2.66	−3.99 ± 0.64
AT4G10603	*SLR1-BP* (S locus-related glycoprotein 1 binding pollen coat protein)	−2.29	−3.15 ± 0.55	−2.64	−2.57 ± 0.57
AT1G29140	Pollen Ole e 1 allergen and extensin family protein	−3.75	−5.85 ± 1.94	−4.76	−6.05 ± 3.63
AT5G45880	Pollen Ole e 1 allergen and extensin family protein	−4.09	−6.26 ± 2.00	−6.70	−5.81 ± 0.82
** *H^+^-ATPase* **					
AT5G57350	*HA3*	−2.12	−2.83 ± 0.74	−2.28	−2.16 ± 0.46
AT2G07560	*HA6*	−2.95	−4.31 ± 1.35	−3.48	−3.87 ± 1.00
AT3G42640	*HA8*	−2.19	−2.50 ± 0.95	−2.74	−2.54 ± 1.49
AT1G80660	*HA9*	−3.84	−4.92 ± 1.00	−4.52	−5.08 ± 1.54
AT3G08560	*VHA-E2*	−2.50	−3.41 ± 0.21	−2.90	−2.94 ± 1.14
** *Protein kinases* **					
AT2G07040	*PRK2A* (Leucine-rich repeat protein kinase)	−5.07	−8.55 ± 4.81	−7.81	−8.47 ± 3.81
AT2G18470	*PERK4* (Proline-rich protein kinase)	−3.24	−4.52 ± 1.24	−4.07	−4.18 ± 0.51
AT2G21480	*BUPS2* (Malectin/receptor-like)	−3.96	−6.52 ± 4.87	−5.45	−7.75 ± 2.38
AT4G39110	*BUPS1* (Malectin/receptor-like)	−3.14	−1.99 ± 0.49	−4.65	−2.10 ± 0.84
AT5G28680	*ANX2* (Malectin/receptor-like)	−5.86	−8.16 ± 6.36	−12.79	−25.23 ± 13.51

^a^: All DEGs selected have FDR < 0.05 and fold changes (FC) great than two fold; ^b^: Gene names are italicized.

**Table 2 cells-12-01542-t002:** RT-qPCR of seven RALF genes and their predicted pI and molecular weight (MW).

Gene_ID ^a^	Gene Description (Clade) ^b^	TC-1 (n = 3)		TC-2 (n = 3)		Predicted pI and MW ^c^	
		RNAseq (FC)	RT-qPCR (FC ± SD)	RNAseq (FC)	RT-qPCR (FC ± SD)	pI	MW (kD)
AT1G28270	*RALF4* (III-B)	−4.27	−7.00 ± 2.79	−5.96	−8.37 ± 4.59	9.76	8.7
AT1G61563	*RALF8* (IV-C)	−3.67	−5.26 ± 1.60	−5.35	−5.66 ± 1.00	9.30	6.6
AT1G61566	*RALF9* (IV-C)	−3.51	−4.77 ± 1.26	−5.27	−5.74 ± 1.38	9.27	6.6
AT2G22055	*RALF15* (IV-C)	−5.73	−2.48 ± 0.39	−23.41	−16.92 ± 2.40	9.76	6.5
AT2G33775	*RALF19* (III-B)	−5.26	−9.10 ± 5.00	−9.69	−8.14 ± 3.43	10.57	7.5
AT3G25165	*RALF25* (IV-B)	−8.24	−2.48 ± 0.67	−23.55	−82.52 ± 18.38	10.08	7.0
AT4G14020	RALF family protein (IV-A)	−3.39	−4.29 ± 1.30	−3.78	−4.02 ± 1.27	10.17	6.8

^a^: All DEGs selected have FDR < 0.05 and fold changes (FC) great than two fold; ^b^: Gene names are italicized; ^c^: Amino acid sequences were retrieved from TAIR (https://www.arabidopsis.org/).

## Data Availability

Arabidopsis Genome Initiative locus identifiers for each gene mentioned in this study are listed in Table 1 and Table 2, Appendix A, and Appendix A. The accession number of *EaF82* is FJ666044. Raw data obtained from RNAseq analysis have been deposited into NCBI’s Gene Expression Omnibus under the accession codes GSE171459 (https://www.ncbi.nlm.nih.gov/geo/) on 5 April 2021.

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
