# Peer review of "A Rapid Alkalinization Factor-like Peptide EaF82 Impairs Tapetum Degeneration during Pollen Development through Induced ATP Deficiency"

_cells, 2023, doi:10.3390/cells12111542_

Round 1

Reviewer 1 Report

In this study, the author introduced that EaF82 is a peptide similar to RALF and exhibits alkaline activity, which demonstrated its interaction with the subunit AKIN10 of SnRK1 kinase. The role of EaF82 was determined to be mediated by AKIN10, leading to abnormal transcription and metabolism, leading to ATP deficiency and damaging pollen development. This article fully demonstrates the mechanism of EaF82 during the development of tapetum. Personally, I believe that this study has great research significance, but before the paper can be accepted for publication, some minor modifications are needed:

1. It is suggested to supplement some functions of genes related to pollen abortion caused by abnormal tapetum,for example,qPCR test;

2. It is recommended to annotate the anther development process diagram in Fig. 3d, indicating the anther structure in the cross-sectional diagram.

3. It is suggested to supplement the PCD process of wild type and TB-2 anther tapetum degradation to fully prove the conclusions obtained.

Suggest conducting a professional English polish throughout the entire text to improve quality

Author Response

Comments and Suggestions for Authors

In this study, the author introduced that EaF82 is a peptide similar to RALF and exhibits alkaline activity, which demonstrated its interaction with the subunit AKIN10 of SnRK1 kinase. The role of EaF82 was determined to be mediated by AKIN10, leading to abnormal transcription and metabolism, leading to ATP deficiency and damaging pollen development. This article fully demonstrates the mechanism of EaF82 during the development of tapetum. Personally, I believe that this study has great research significance, but before the paper can be accepted for publication, some minor modifications are needed:

Response: We thank the reviewer for the positive comments on our study. We really appreciate it.

  1. It is suggested to supplement some functions of genes related to pollen abortion caused by abnormal tapetum, for example, qPCR test;

 Response: We appreciate the reviewer’s suggestion. It is a challenge to identify the specific genes responsible for pollen abortion. Nevertheless, we selected nine pollen-related genes (AT1G19890, AT1G19960, AT1G21000, AT1G35490, AT1G24520, AT5G17480, AT4G10603, AT1G29140, and AT5G45880) and used qPCR to demonstrate that their expression levels indeed were reduced in the TC lines, consistent with the results observed in the RNAseq analysis (Table 1). Furthermore, a group of RALFs (AT1G61566, AT1G61563, AT1G28270, AT2G33775, and AT3G25165), which were previously reported to be commonly affected in four (DYT1-TDF1-AMS-MS188) tapetum mutants (Reference #12, Li et al., 2017), were also found in our down-regulated DEG list. Their expression levels were further quantified using qPCR. Both the RNAseq and qPCR results are consistent (see Table 1), providing support for the observed abnormality of the tapetum.   

  1. It is recommended to annotate the anther development process diagram in Fig. 3d, indicating the anther structure in the cross-sectional diagram.

Response: We have annotated T: tapetum, Tds: tetrads, E: epidermis, En: endothecium, MSp: microspores, PG: pollen grain onto the Fig3d, and have also replaced the figure legend correspondingly. 

  1. It is suggested to supplement the PCD process of wild type and TB-2 anther tapetum degradation to fully prove the conclusions obtained.

Response: We fully understand that the suggested study is important and can strongly support our conclusions. The immunohistochemistry analysis can be used to systemically investigate the PCD process between wild type and TB-2 anther tapetum degradation; however, it requires significant efforts.  Since our current manuscript has already contained large number of observations and data, the suggested experiment will be carried out in our future study.

Comments on the Quality of English Language:

Suggest conducting a professional English polish throughout the entire text to improve quality

Response: We appreciate the reviewer’s comment. As suggested, we have polished the manuscript and asked a native English speaker to read and edit it. Please refer to the attached version with track changes to see the revisions made.

Reviewer 2 Report

This paper describes a small cysteine-rich protein found in Golden Pothos that may be important for tapetal cell function. The authors hypothesize that the small protein functions similarly to a RALF protein and may be part of the regulatory mechanism that leads to tapetum degeneration. The manuscript may be significant because it addresses the death of a tissue that is critical for pollen development and thus plant reproduction.  

1. One point that puzzles me is the authors' claim that golden pothos does not flower. I am not familiar with all the species in this genus, so my question is why study a protein involved in tapetum degeneration if the plants do not flower. So, the authors could be saying that the species flowers irregularly or infrequently. Otherwise, the significance of studying a peptide involved in tapetum degeneration in a species that does not flower is diminished. 

2. In the introduction, the authors state that Arabidopsis and tobacco have no homologues of EaF82, but the peptide appears to function in both species. So, while the specific peptide is missing, the interacting proteins should be present in both species. So how do we reconcile these two sets of results? What effect will this have on the Y2H assay? 

3. The authors seem to have re-analyzed the relationships between EaF82 and putative Arabidopsis homologs in the first section of the results and found that the peptide shares some sequence similarities with AtRALF8/9/15. So, this point needs to be clarified, because on the one hand it is reported that there is no homology with Arabidopsis, but on the other hand the authors have provided additional data, so it needs to be made clear why these new data contradict the previous ones. 

4. Other than the previous points, I found the introduction to be professionally written and clear. It covers all the important points.  

5. The legend of Figure 1A states that a specific sequence is used for antibody production. This is the first time that the use of antibodies has been reported. Therefore, it needs more context. Since the methods are at the end of the manuscript in this journal, it is important to say a bit more about the antibody in advance.  

6. The immunoblotting in Figure 2D is interesting, but it lacks a control, the anthers of the TA line. If the peptide is highly expressed in anthers but not in pollen, immunoblotting against anther proteins should show a consistent signal that is at least significantly higher than that of pollen alone. 

7. The data in Figure 2 indicate that when EaF82 expression is driven by its own promoter, silique development is reduced, but pollen is likely to be produced and, although less abundant, is still viable. When compared to the data in Figure 3, EaF82 overexpression driven by the 35S promoter has a strong effect on silique development. The authors emphasize the absence of major effects on the tapetum at S11 in the TB lines. Could the same analysis be performed in TA lines? A histological analysis of TA lines during pollen development and tapetum degeneration could be important. 

8. The 35S promoter is suitable to study the overexpression of EaF82. However, why not test the effect of using a pollen-specific promoter to evaluate the potential effect on pollen (if it exists)? 

9. The data on transgenic tobacco plants are interesting and confirm that EaF82, when overexpressed by the 35S promoter, indeed induces strong effects on flower development and seed set. Why not express EaF82 under the control of its own promoter as done in Arabidopsis? Did the authors check this? This should be an important control. 

10. The analysis of DEGs after overexpression of EaF82 is important and shows that most of the DEGs are related to cell wall modifications and pH changes. This is indeed consistent with the authors' hypothesis that EaF82 is involved in the regulation of cell wall structure in tapetum cells. However, monitoring cell wall changes may be important to test the authors' hypothesis. For example, tapetum cells could be stained with fluorescent dyes or labeled with antibodies against cell wall polysaccharides. This could add value to the paper. Monitoring pH changes in the cell wall could be challenging, but evaluation of cell wall structure is feasible. Even the localization of AGPs could be tested by labeling with commercially available antibodies. 

11. The manuscript is well-written, but I recommend that the authors go over the Results section again because it contains several sentences that sound more like discussion. Because the number of observations and data is large, streamlining the Results section would improve readability and avoid repetition of some sentences. Indeed, the use of several citations in the Results section indicates the presence of discussion-like sentences. 

12. One point needs to be clarified. The authors speculate in the discussion that EaF82 interacts with a cell membrane receptor, is internalized, and then interacts with AKIN10. My question is, why didn't the putative receptor show up in the Y2H assay? Could it be one of the proteins that the authors overlooked? 

My final comment is that the manuscript is a very interesting research topic, and it appears to contribute significantly to describing the events that occur during tapetum degeneration. However, some details must be clarified. 

Round 2

Reviewer 2 Report

I have reviewed this manuscript again and read the authors' response to my previous comments, and I found that most of my previous questions have been resolved. However, I think there are still some points that need to be clarified by the authors: 

Regarding point 1, I understand the author's response. However, this should be more specified in the manuscript, because the current version of the manuscript still reports the simple statement that plants do not flower. So, I think the authors should describe in more detail why they looked at this protein involved in tapetum degeneration in a species that does not flower. So, I suggest they add more information on this point using the text they provided me as a response. 

Regarding point 6, I understand that the authors observed a protein fused to GFP in the anthers and considered this data as an alternative to the immunoblot. But the immunoblot data is missing an important piece. So, I insist to the authors that they should provide a complete blot panel of data. 

I do not agree with the authors' response in point 10. I think that the reviewers should evaluate the quantity and quality of the data, and this is not the responsibility of the authors. They should provide information and the reviewers should decide on the quantity of information. I think that staining the cell wall polysaccharides with fluorescent dyes is easy and can be done without much effort. So, I insist on this information. 

All the other points are OK for me. 

Round 3

Reviewer 2 Report

Dear colleague, I found that you adequately responded to my previous requests. I understand the reason why it is not possible for you to perform the required experiments. Anyway please consider to include even more basic test in future works. They can provide some useful insights.